# Influence of Hydrogenation on Morphology, Chemical Structure and Photocatalytic Efficiency of Graphitic Carbon Nitride

**DOI:** 10.3390/ijms222313096

**Published:** 2021-12-03

**Authors:** Daria Baranowska, Tomasz Kędzierski, Małgorzata Aleksandrzak, Ewa Mijowska, Beata Zielińska

**Affiliations:** Department of Nanomaterials Physicochemistry, Faculty of Chemical Technology and Engineering, West Pomeranian University of Technology, Piastow Ave. 42, 71-065 Szczecin, Poland; tomasz.kedzierski@zut.edu.pl (T.K.); malgorzata.wojtoniszak@zut.edu.pl (M.A.); ewa.borowiak-palen@zut.edu.pl (E.M.)

**Keywords:** graphitic carbon nitride, thermal treatment, hydrogen atmosphere, RhB decomposition

## Abstract

In this contribution, the effect of hydrogenation conditions atmosphere (temperature and time) on physicochemical properties and photocatalytic efficiency of graphitic carbon nitride (g-C_3_N_4_, gCN) was studied in great details. The changes in the morphology, chemical structure, optical and electrochemical properties were carefully investigated. Interestingly, the as-modified samples exhibited boosted photocatalytic degradation of Rhodamine B (RhB) with the assistance of visible light irradiation. Among modified gCN, the sample annealed at 500 °C for 4 h (500-4) in H_2_ atmosphere exhibited the highest photocatalytic activity—1.76 times higher compared to pristine gCN. Additionally, this sample presented high stability and durability after four cycles. It was noticed that treating gCN with hydrogen at elevated temperatures caused the creation of nitrogen vacancies on gCN surfaces acting as highly active sites enhancing the specific surface area and improving the mobility of photogenerated charge carriers leading to accelerating the photocatalytic activity. Therefore, it is believed that detailed optimization of thermal treatment in a hydrogen atmosphere is a facile approach to boost the photoactivity of gCN.

## 1. Introduction

The pioneering work of g-C_3_N_4_ as a semiconductor for photocatalytic water splitting was first reported by Wang et al. in 2009 [1,2]. Since that time, g-C_3_N_4_ has attracted extensive attention in material engineering induced by its winsome electronic structure, i.e., having a suitable energy bandgap (*Eg*) and being a metal-free conjugated indirect semiconductor. First-principle calculations anticipated seven phases of C_3_N_4_, namely α-C_3_N_4_, β-C_3_N_4_, cubic C_3_N_4_, pseudocubic C_3_N_4_, g-h-triazine, g-o-triazine, and g-h-heptazine with an *Eg* of 5.49, 4.85, 4.30, 4.13, 2.97, 0.93, and 2.88 eV, respectively [1,3]. At ambient conditions, g-C_3_N_4_ is considered the most stable allotrope. g-C_3_N_4_ is a 2D material in which the stacking layers (covalent C—N bonds) are held by van der Waals forces. Each layer of g-C_3_N_4_ is composed of tri-s-triazine units linked by amino groups [1,4]. The tri-s-triazine ring structure provides (1) thermal stability (up to 600 °C in the air) and (2) chemical stability (both in acidic and alkaline environments) [1,3]. The aforementioned properties make g-C_3_N_4_ a promising material for a variety of photocatalytic applications exploiting solar energy. However, its efficiency is greatly hampered by slow charge carrier mobility and the high probability of their recombination [5]. The commonly used strategy for overcoming these issues includes introducing surface defects by controlling the preparation parameters (e.g., careful selection of polymerization temperature, post-treatment in the reductive or inert atmospheres, etc.) [6]. 

To be more specific, the utilization of both hydrogen atmosphere and high temperature causes delamination of carbon nitride and introduces defects on its surface, namely nitrogen vacancies [5]. These vacancies can play a crucial role in modifying photocatalysts by introducing additional energy levels or active sites enhancing photocatalytic performance [7,8]. Contrary to the detailed studies on oxygen vacancies [7], the underlying role of nitrogen vacancies in boosting photocatalytic efficiency is rarely considered [7,9]. 

There are only a few reports on the effects of nitrogen vacancies on the g-C_3_N_4_ surface. However, the potential of as-modified materials has not been fully revealed yet. g-C_3_N_4_ with nitrogen vacancies was prepared via post-treatment in a hydrogen atmosphere at various temperatures [5,6,7]. For example, Niu et al. [7] heated the g-C_3_N_4_ in the H_2_ atmosphere. Compared to the reference, the as-modified g-C_3_N_4_ has an extended visible light absorption range and suppressed recombination photogenerated charge carriers caused by the presence of nitrogen vacancies. Consequently, the as-modified sample shows enhanced photocatalytic activity than the reference in both •OH radicals generation and Rhodamine B degradation under UV and visible light illumination. The authors revealed that the modified sample is around two times more active in •OH radicals generation as the reference. 

A similar dependence is observed in the case of photocatalytic degradation of RhB. Furthermore, the sample with nitrogen vacancies exhibits no significant changes after both photocatalytic reactions, which indicates good stability of the as-modified photocatalyst. Kang et al. [6] modified g-C_3_N_4_ by the thermal treatment process at 400 °C in an H_2_ atmosphere under high pressure. Among the hydrogen-treated samples, the gCN(6h) sample presents the highest H_2_ evolution rate (270 µmol^−1^ h^−1^), 2.3 times higher than the reference. The improvement is attributed to the following effects: (1) widened visible light absorption range, (2) the increased specific surface area, and (3) formed pores for shortening the diffusion lengths of the photogenerated charge carriers to reach the surface. Li et al. [5] prepared exfoliated g-C_3_N_4_ by high-temperature H_2_ treatment. The results revealed that the H_2_ treatment of g-C_3_N_4_ for 2 h provides a superior photocatalytic H_2_ evolution rate of 4.8 µmol h^−1^. This H_2_ evolution rate represents ~7 times improvement over reference. These results revealed that the disorder caused by the H_2_ treatment serves as a good proxy for photocatalytic activity. 

Moreover, the effects of the precursor and preparation conditions on the photocatalytic performance of g-C_3_N_4_ were also studied [10,11]. For example, Tay et al. [10] showed enhanced visible light absorption and photocatalytic H_2_ evolution by introducing nitrogen vacancies in g-C_3_N_4_ produced by direct thermal polymerization of dicyandiamide in H_2_ atmosphere. For comparison, g-C_3_N_4_ samples were calcinated in the different atmospheres (air, N_2_, forming gas (5% H_2_ + 95% Ar) and pure H_2_) with constant temperature and treatment duration (550 °C for 4 h, 5 °C min^−1^). g-C_3_N_4_(H_2_) shows ~5 times higher H_2_ evolution with the assistance of visible light illumination compared to g-C_3_N_4_(air), confirming the importance of gas atmosphere during catalyst preparation. Zhang et al. [11] studied the effect of precursor type, thermal polymerization temperature, and the atmosphere to achieve a highly effective catalyst based on g-C_3_N_4_. g-C_3_N_4_ prepared from ammonium thiocyanate (AT, NH_4_SCN) as precursor at 550 °C for 5 h under H_2_ atmosphere showed the highest activity. The AT-H2-550-5 can catalyze the complete degradation of RhB (C_0_ = 10 mg L^−1^) after 2 h of blue light illumination. The prepared sample showed good stability and reusability after four cycles. Evidence shows that controlling the effects of preparation conditions has played a crucial role in lowering the charge carrier recombination probability by shortening the diffusion length for boosted photocatalytic efficiency [5,6,11,12]. Therefore, the synergistic controlling of the band structure and the microstructure of g-C_3_N_4_ is highly desirable but challenging [6].

The potential of g-C_3_N_4_ modified by heat-treatment conditions in a hydrogen atmosphere has not been fully described yet. Therefore, the effect of temperature and duration of annealing in the H_2_ atmosphere on physicochemical properties and photocatalytic efficiency of graphitic carbon nitride has been investigated. A detailed analysis of morphology, chemical structure, optical and electrochemical properties of as-modified materials has been provided in the present contribution. The as-modified samples were examined in Rhodamine B (RhB) decomposition reaction with the assistance of visible light. 

## 2. Results and Discussion

### 2.1. Transmission Electron Microscopy 

Appendix A (see Appendix A) presents TEM images of pristine gCN, 400-2, 450-2, 500-1, 500-2, 500-3, 500-4 and 550-2. Each sample displayed a typical morphology of 2D and platelet-like objects with a tendency to fold and aggregate. The morphology of reference gCN shows bulk material with a non-porous structure [6]. TEM characterization shows that H_2_ treatment leads to the exfoliation of all modified catalysts. Deeper magnification shows that the pores are randomly distributed on the modified carbon nitrides surface. Moreover, annealing in H_2_ atmosphere caused a significant decrease in thickness and lateral size of the obtained materials, confirmed by further AFM analysis. 

### 2.2. Atomic Force Microscopy

AFM images with corresponding height profiles of reference gCN, 500-2, and 500-4 are illustrated in Figure 1. Additionally, Appendix A displays AFM images of 400-2, 450-2, 500-1, 500-3, and 550-2. The thickness and lateral size of gCN flakes ranged from 4–9 nm and from 0.067–0.136 µm, respectively. The aforementioned parameters were reduced after annealing in H_2_ atmosphere. The thickness of 500-2 is in the range of 1–5 nm, with a lateral size of 0.031–0.133 µm. The thickness and lateral size of 500-4 are 1–5 nm and 0.036–0.098 µm, respectively. The obtained 500-2 and 500-4 are composed of 3–15 atomic layers when the reference gCN is composed of 12–28 atomic layers [13]. The characteristic parameters, such as thickness, lateral size, and the number of layers of the obtained samples, are summarized in Appendix A. It confirms that longer time of H_2_ treatment did not lead to significant changes in the reduction of thickness and lateral size of the samples in contrast to the elevated temperature. Furthermore, annealing in H_2_ atmosphere causes the hydrogenation of tri-s-triazine units forming dangling N–H moieties, breaking the weak van der Waals forces between layers, and causing the controlled exfoliation of bulk gCN [5]. AFM analysis proves that the temperature of annealing in H_2_ atmosphere has a significant effect on the carbon nitrides thickness and a moderate impact on their lateral size.

### 2.3. X-ray Powder Diffraction

XRD patterns of studied samples are presented in Figure 2. All materials displayed a similar crystalline phase with two characteristic reflections at approximately 13° and 27° assigned to the (100) and (002) planes, respectively (JCPDS: 01-087-1526). The aforementioned reflections have been attributed to the in-plane packing motif of tri-s-triazine units and the interplane periodic stacking of the layers along the c-axis, respectively [5,6,14]. The comparison of XRD data shows position shifts and changed intensity. More specifically, the (002) diffraction peak is shifted from 27.33° for reference gCN to 27.44° for 500-2, implying structural changes caused by annealing in the H_2_ atmosphere. The shifts to higher angles while treatment with higher temperature cause the interlayer spacing to decrease. Similar dependence shifts are commonly reported in state of the art with g-C_3_N_4_ prepared at higher temperatures [5,15,16]. These observations are consistent with the decrease in the interlayer distance that occurred due to the further condensation induced through the additional thermal treatment of the gCN, resulting in a more stable structure within the layers [5]. 

These changes can be explained for two reasons: (1) the thermal treatment favored the increased crystallinity by regulating the atomic long-range ordering towards a better packing of the building block sheets; (2) a reductive atmosphere caused partial loss of nitrogen atoms from the hydrogen-treated samples framework so that the in-plane long-range ordering of the atoms was destroyed [6,17,18]. 

### 2.4. Fourier Transform Infrared Spectroscopy 

The chemical structure of prepared materials was examined using FT-IR spectroscopy (Figure 3). All materials displayed characteristic spectra of graphitic carbon nitride. Moreover, all samples expose almost the same position of the vibration modes, indicating a similar molecular structure. The absorption at 810 cm^−1^ is attributed to the heptazine rings. The absorptions in the range of 1100–1700 cm^−1^ have been assigned to the stretching modes of C–N and C = N [19,20,21]. What is more, the vibrations between 2800–3400 cm^−1^ expected from the hydrogenation of carbon nitride were detected in the IR spectra for all modified samples. They are associated with the absorbed hydroxyl groups/water molecules and uncondensed amine species NH/NH_2_ [6]. 

### 2.5. Specific Surface Area Measurement 

Nitrogen adsorption–desorption isotherms of studied samples are shown in Figure 4a,b. The isotherms curves of materials display the typical type IV isotherms and type H4 hysteresis loops, proving the formation of mesopores in the obtained samples. Each isotherm does not have clear saturated adsorption platforms, which indicates a very irregular pore structure [11]. The results revealed that annealing in the H_2_ atmosphere caused an increase in specific surface area (SSA), total pore volume, and average pore diameter. Table 1 contains the data of the aforementioned parameters with corresponding energy band gaps of all studied samples. The carbon nitrides reduced in the hydrogen atmosphere have a higher value of all three measures than reference gCN. The SSA of gCN was 8.89 m^2^ g^−1^, whereas 500-2 and 500-4 exhibited 28.69 and 24.11 m^2^ g^−1^, accordingly. The total pore volume of gCN was 0.003 cm^3^ g^−1^. For 500-2 and 500-4, the values of total pore volume increased to 0.013 and 0.012 cm^3^ g^−1^, respectively.

Similarly, the average pore diameter increased from 0.88 nm for gCN to 1.03 and 1.09 for 500-2 and 500-4, accordingly. Interestingly, 400-2 has the largest value of SSA but exhibits a significantly lower effectivity in photocatalytic RhB decomposition reaction. Inferencing, the photocatalytic activity of carbon nitride is mainly caused by the inherent electronic properties of a particular sample and not only its SSA [5,22]. 

### 2.6. X-ray Photoelectron Spectroscopy 

The chemical composition and relative atomic percentages of the pristine gCN and 500-4 (the sample with the highest photocatalytic activity) were analyzed by XPS. The spectra revealed that the studied materials are composed of carbon, nitrogen, and oxygen [23]. The atomic concentration of these elements was listed in Appendix A. The obtained results indicate that for 500-4, the amount of nitrogen increased, whereas the amount of carbon and oxygen decreased. The detailed analysis of the individual components was calculated using the peak-fitting procedure to the C 1s and N 1s spectra of both samples and presented in Figure 5 and Appendix A. Peaks located at 288 and 399 eV are assigned to C 1s and N 1s signals, respectively. The C 1s region includes three peaks. Specifically, the peak at 284 eV belongs to a special sp^2^ C–C bond in the CN from g-C_3_N_4_ [11]. The peak at 286.5 eV is attributed to the C atom on the aromatic ring connected to NH_x_ [11]. The strong peak at 288 eV can be assigned to the sp^2^ hybrid C atom connected to N in the triazine ring N–C = N [11]. The N 1s region consists of three contributions which are associated with N–C = N (N_2_C), N–H_x_, and C–N_3_ (N_3_C) [19]. The peak at 399 eV can be assigned to the sp^2^ hybrid nitrogen orbital in the triazine ring (N_2_C) [11]. The peak at 400.5 eV is assigned to bridging N atoms–the sp^2^ hybrid N binds with three C atoms (N_3_C) [11]. The peak observed at 401.5 eV corresponds to residual amino groups (N–H_2_) or sp^3^ hybrid nitrogen of the amino groups (C–N–H) [11], commonly labeled as N–H_x_. It can be observed that because of nitrogen vacancies presence, the content of the following bonding increases: C–NH_x_ (from 8.40 to 21.66), N_2_C (from 81.02 to 88.41) and N–H_x_ (from 4.04 to 9.22) compared to pristine gCN. Besides, the amount of N–C = N, C–C/C = C, and N_3_C appropriately decreases. The XPS analysis indicates the creation of nitrogen vacancies on modified 500-4 act as additional active sites on its surface, accelerating photocatalytic activity. 

### 2.7. Diffuse Reflectance Spectroscopy 

The UV-vis DRS absorption spectra of studied materials are displayed in Figure 6. To establish the value of the energy bandgap (*Eg)*, the Tauc method based on DR spectra was applied. The band gaps are 2.63, 2.65, 2.68, 2.69, 2.70, 2.71, 2.75 and 2.80 eV for gCN, 400-2, 450-2, 500-1, 500-2, 500-3, 500-4 and 550-2, respectively. All the hydrogen-treated samples have broadened energy bandgap compared to pristine gCN. The value of *Eg* rises with the increase in temperature and duration treatment in H_2_ atmosphere. 

### 2.8. Photoluminescence

Figure 7 presents the photoluminescence spectra of the studied samples. The reference gCN exhibits a strong emission peak centered at 450 nm, resulting from the fast recombination of photogenerated charge carriers [6]. All modified materials show notably extinct PL peaks with the increase in the temperature and duration of treatment. The PL spectrum of 500-4 shows the lowest PL intensity as compared to reference gCN, indicating the lowest rate of photogenerated carrier recombination, which is beneficial for photocatalytic performance. Broadened optical absorption in the visible range, higher specific surface area, and lower recombination rate of the hydrogen-treated carbon nitrides suggest boosting photocatalytic activity [10]. 

### 2.9. Electrochemistry

The H_2_ treated carbon nitrides provide electron donors (the introduction of hydrogen to the g-C_3_N_4_ can be interpreted as reductive or n-type doping), introducing defects on its surfaces, therefore improving the carrier mobility of as-modified materials [5,22,24]. Considering temperature dependence (Figure 8a,b), it was noticed that all modified samples show improved photocurrent response, and the semicircles are smaller, which is desirable in photocatalytic performance. To be more specific, the highest photocurrent response is achieved for 450-2, which is 2.68 times higher than pristine gCN. An analogy is observed in the case of time dependence (Figure 8c,d). Here, the highest photocurrent response is achieved for 500-4 (3.14 times higher than reference gCN). Consequently, the photogenerated carriers are better transferred and separated on the surfaces of modified carbon nitrides intersperse to higher photocatalytic activity. Hence, annealing of graphitic carbon nitride in the hydrogen atmosphere results in different chemical structures, surface area and porosity, optical and electrochemical properties, which all will affect the photocatalytic activity [10]. 

### 2.10. Photocatalytic Performance 

The photocatalytic efficiency of the studied catalysts was investigated in the RhB degradation process under visible light illumination (Figure 9). RhB without the addition of a catalyst displayed irrelevant self-decomposition with the assistance of visible light irradiation. All modified samples show gradually enhanced photocatalytic activity. As shown in Figure 9a, a higher temperature of annealing in H_2_ atmosphere up to 500 °C led to an increase in the efficiency of decomposition of RhB. An increase in the temperature of H_2_ treatment up to 550 °C decreased the removal efficiency of RhB. This is due to the much higher value of bandgap energy (2.81 eV), which influenced a decrease in optical and electrochemical properties of 550-2. The data collected from photoluminescence, chronoamperometry, and electrochemical impedance spectroscopy proves that the transfer and separation of photogenerated carriers are hampered in comparison to sample with the highest photoactivity (500-4) in the RhB degradation process. Among the temperature-dependent samples, 500-2 has the highest efficiency, 1.27 times higher than that of pristine gCN. In the case of time-dependent samples (Figure 9b), 500-4 catalyzed the complete decomposition of dye after 1.5 h of irradiation (1.76 times higher compared to gCN). Additionally, the histograms of photocatalytic effectivity are presented in Figure 9c,d. The boosted photocatalytic activity was explained by enhanced specific surface area, improved transfer, and separation of photogenerated carriers, successful exfoliation, and nitrogen vacancies presence achieved during H_2_ treatment [6]. Finally, it is revealed that the changing trend of photocatalytic RhB decomposition of samples prepared under different conditions is in good agreement with their photoluminescence spectra and electrochemical measurements. Additionally, the first-order constants *(k)* for photocatalytic RhB degradation over modified materials were calculated by *ln(C/C_0_) = -kt*. The time-course variations of *ln(C/C_0_)* of the samples different versus temperature were shown in Figure 9e and versus time in Figure 9f. The kinetic constants *(k)* with their *R^2^* are listed in Appendix A. The obtained results showed that the time of annealing in H_2_ has a crucial impact on photocatalytic efficiency.

The stability of 500-4, the sample with the highest photoactivity, was tested in four parallel cycles. As shown in Appendix A, 500-4 was found to be photochemically stable under visible illumination. Approximately 15% of RhB is decomposed after four runs indicating good stability of as-prepared material. The chemical structure of the catalyst after photocatalytic reaction (red lines) was compared to data before photocatalysis (black lines) and presented in Appendix A. The XRD patterns and FTIR spectra do not show any significant changes on its surface or molecular structure, confirming the good durability of 500-4. 

Generally, the degradation of organic dyes with the use of photocatalysts is mainly determined by the separation and recombination rates of the photogenerated carriers on the catalyst surface. A critical role in photocatalytic activity is attributed to the presence of nitrogen vacancies achieved by the annealing of graphitic carbon nitride in a hydrogen atmosphere. The incorporation of nitrogen vacancies in 500-4 separates the photogenerated electron-hole pairs and generates additional photodegradation sites. To propose a possible mechanism of dye decomposition, the value of valance bands of gCN and 500-4 were calculated using the VB XPS technique. VB XPS spectra were presented in Appendix A. The value of valance bands of gCN and 500-4 were established to be 3.65 and 3.15, respectively.

Furthermore, these samples’ conduction bands (CB) positions were calculated from the formula *E_CB_ = Eg* − *E_VB_*. The value of conduction bands of gCN and 500-4 were calculated to be 1.02 and 0.40, accordingly. The energy band structures of gCN and 500-4 are shown in Figure 10. Both the VB and CB of 500-4 are located higher in comparison to the reference gCN. Additionally, the CB value of 500-4 is more negative than that of E^0^ (redox potential of O_2_/•O_2_^−^ = −0.046 eV vs. NHE) due to the conduction band potential, resulting in easy transformation of dissolved oxygen molecules into superoxide anion radicals (O_2_^−^) to response easily to RhB dye molecules, which are the main active species of photodegradation of RhB dye. Additionally, the VB potential of 500-4 is not more positive than that of E^0^ (•OH/OH^−^ = 1.99 eV vs. NHE); thus the holes (h^+^) could not oxidize OH^−^/H_2_O to generate hydroxyl radicals (•OH) directly in 500-4 photocatalytic process [25]. However, •OH can be produced from the reaction between •O_2_^−^ and H_2_O_2_, accelerating the photocatalytic activity [26]. Moreover, the ability of the organic dye itself decomposition should be considered [26]. In this case, the dye molecule can provide the electron to 500-4 through the irradiation process. Cationic RhB molecules are adsorbed on the surface of the catalyst. Then, the electrons of organic dye are excited under the visible light to generate the excited electrons starting the photocatalytic reactions [27]. 

## 3. Materials and Methods

### 3.1. Materials

#### 3.1.1. Preparation of gCN

Melamine (C_3_H_6_N_6_) was acquired from Sigma Aldrich. 10 g of melamine powder was placed in crucible with a lid. Next, it was heated in the muffle furnace at 550 °C for 4 h with a heating rate of 2 °C min^−1^ under air conditions. The yellow powder was obtained and labeled as gCN.

#### 3.1.2. Preparation of Modified gCN by Annealing in H_2_ Atmosphere

A certain amount of gCN was annealed in the furnace at a temperature range between 400 and 550 °C in the different duration range between 1 and 4 h under the H_2_/N_2_ gas mixture in the ratio of 1/19, respectively and flow of 20 mL min^−1^. The as-prepared products were labeled as T-t (T—the temperature and t—the thermal treatment duration). Detailed conditions of the preparation of T-t materials are presented in Table 2. The hydrogen treatment of gCN performed at 400 °C for 2 h (sample 400-2) led to a product yield of ~50 wt%, far higher than that for 550-2 (~15 wt%). Further increase in temperature or duration treatment caused negligible yield (below 5 wt%). 

### 3.2. Methods

The morphology of studied samples was determined using a Transmission Electron Microscope (TEM) FEI Tecnai F20 with 200 kV voltage. Atomic Force Microscope (AFM) analysis was performed using Nanoscope V Multimode 8. X-ray Powder Diffraction (XRD) patterns of studied materials were recorded on Aeris, Malvern Panalytical employing CuKα radiation. X-ray Photoelectron Spectroscopy (XPS) was conducted using a Prevac system with a Scienta SES 2002 electron energy analyzer employing MgKα radiation. The analysis chamber was evacuated to a pressure below 5 × 10^−9^ mbar. Fourier Transform Infrared Reflection (FTIR) spectra were acquired on Nicolet 6700 FT-IR spectrometer. The specific surface area with pore size distribution was measured using N_2_ adsorption/desorption isotherm by a Micromeritics ASAP 2460 apparatus. The optical absorption spectra of samples were tested in a UV-vis spectrophotometer JASCO V-770 in the diffuse reflectance mode (DRS). Photoluminescence (PL) spectroscopy was performed with 280 nm excitation at room temperature using F-7000, Hitachi, (Tokyo, Japan). 

The electrochemical measurements were determined with a three-electrode test cell coupled with a potentiostat (Autolab PGSTAT302N). The counter electrode (platinum plate), reference electrode (calomel electrode), and inorganic electrolyte (0.1 M Na_2_SO_4_ solution) were used. The working electrode was achieved using 2 mg of catalyst sonicated in ethanol:water solution (volume ratio 1:3) and 25 µL of Nafion solution (5 wt%, Sigma Aldrich, St. Louis, MO, USA) for 1 h. A total of 50 µL of the obtained solution was coated onto the FTO glass side (Sigma Aldrich). The photocurrent response (Chronoamperometry) test was conducted at 0.5 V vs. SCE, while the Electrochemical Impedance Spectroscopy test was measured at 0.15 V vs. SCE. 

The photocatalytic performance of obtained materials was examined in Rhodamine B (RhB) decomposition reaction with the assistance of visible light irradiation (150 W halogen lamp with hot-mirror cut-off light filter, Photon Institute, Warsaw, Poland). The catalyst (50 mg) was dispersed into an aqueous RhB solution (100 mL, 5 mg dm^−3^) in an open glass reactor. Visible-light irradiation was preceded by stirring the sample in the darkness for 1 h to establish the adsorption–desorption equilibrium. Afterward, the solution was irradiated for 5 h. The solution was taken at equal time intervals (2 mL) and the absorbance was determined by UV-vis spectrophotometer. The decomposition efficiency of organic dye was calculated according to the equation: C/C_0,_ where C_0_ and C stand the initial concentration of dye solution and concentration after a predetermined irradiation time, respectively. Additionally, photocatalytic experiments were carried out in triplicate, and standard errors were calculated. For the stability test of 500-4, four experiments of RhB decomposition were performed. After each cycle, the catalyst was recovered and purified by washing with distilled water and ethanol and dried under vacuum for 24 h. All experiments were conducted under the same conditions.

## 4. Conclusions

In summary, the effect of hydrogenation (temperature and duration dependence) on physicochemical properties and photocatalytic efficiency of graphitic carbon nitride was studied in great detail. The as-modified carbon nitrides boosted photocatalytic dye decomposition with the assistance of visible light. It was noticed that treating gCN with hydrogen at elevated temperatures caused the creation of nitrogen vacancies on gCN surfaces, which act as highly active sites enhancing the photocatalytic efficiency. Among modified gCN, the sample annealed at 500 °C for 4 h in H_2_ atmosphere (500-4) exhibited the highest photocatalytic efficiency, which is 1.76 times higher compared to reference gCN. The improvement of 500-4 photo efficiency can be understood as the common impact of the following parameters: reduction of a thickness (from 6.46 to 2.28), increasing specific surface area (from 8.89 to 24.11), reduction of PL intensity and enhancement of photoresponse. The boosted photocatalytic activity was explained by enhanced specific surface area, improved mobility of photogenerated carriers, successful exfoliation, and nitrogen vacancies presence achieved during H_2_ treatment. 

## Figures and Tables

**Figure 1 ijms-22-13096-f001:**
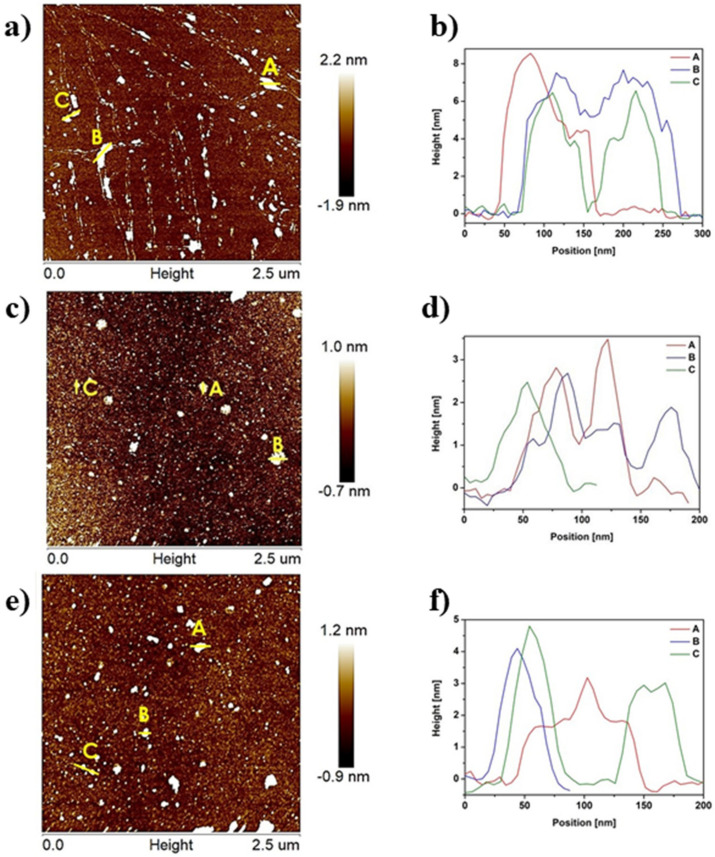
AFM images with corresponding height profiles of gCN (**a**,**b**), 500-2 (**c**,**d**), and 500-4 (**e**,**f**).

**Figure 2 ijms-22-13096-f002:**
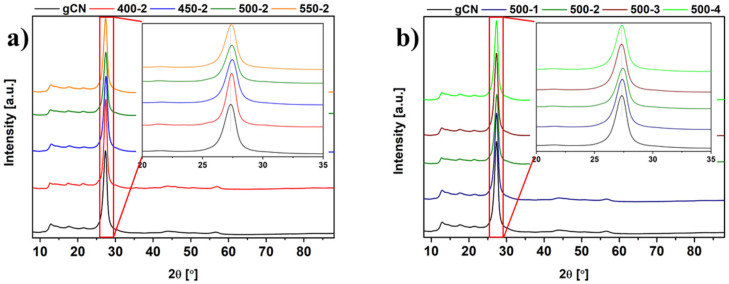
XRD patterns of gCN prepared under different conditions: temperature dependence (**a**), and time dependence (**b**).

**Figure 3 ijms-22-13096-f003:**
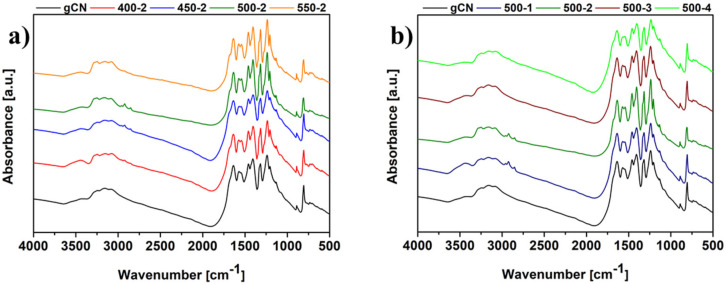
FTIR spectra of gCN prepared under different conditions: temperature dependence (**a**), and time dependence (**b**).

**Figure 4 ijms-22-13096-f004:**
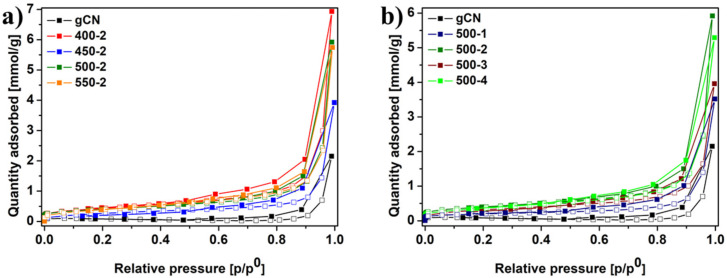
Nitrogen adsorption–desorption isotherms of gCN prepared under different conditions: temperature dependence (**a**), and time dependence (**b**).

**Figure 5 ijms-22-13096-f005:**
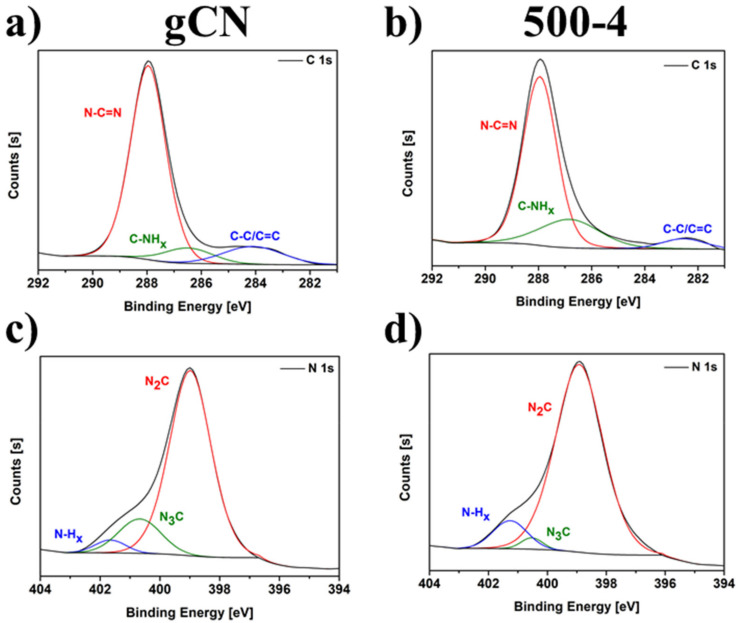
C 1s and N 1s XPS spectra of gCN (**a**,**c**), and 500-4 (**b**,**d**).

**Figure 6 ijms-22-13096-f006:**
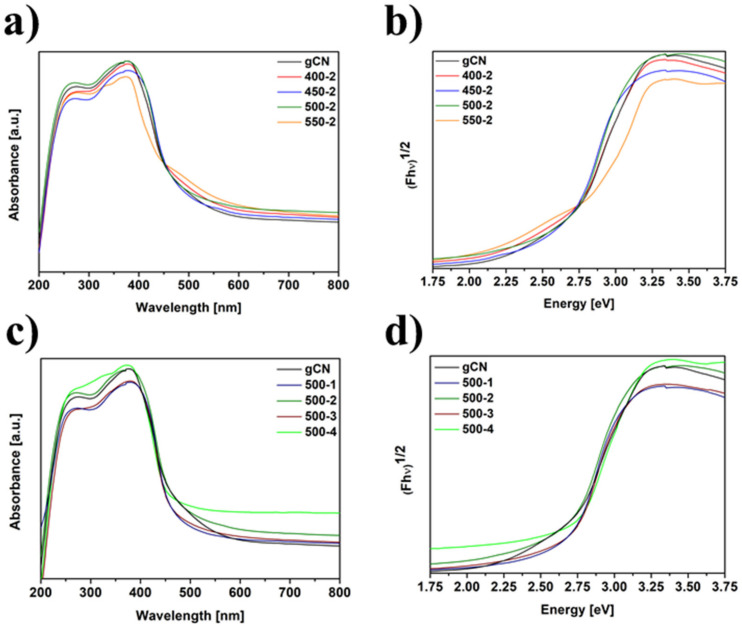
DR/UV-vis spectra of gCN prepared at different temperatures (**a**), Tauc plot of gCN prepared at different temperatures (**b**), DR/UV-vis spectra of gCN prepared at different times (**c**), Tauc plot of gCN prepared at different times (**d**).

**Figure 7 ijms-22-13096-f007:**
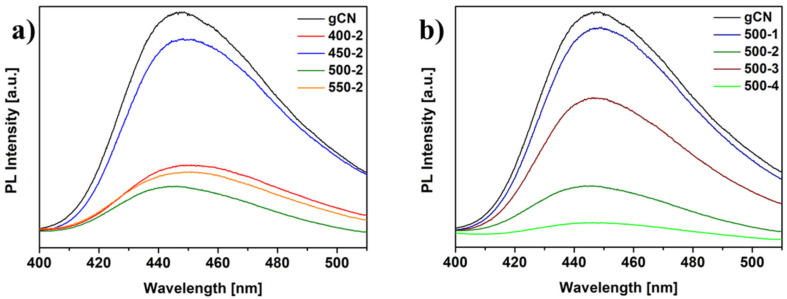
PL of gCN prepared under different conditions: temperature dependence (**a**) and time dependence (**b**).

**Figure 8 ijms-22-13096-f008:**
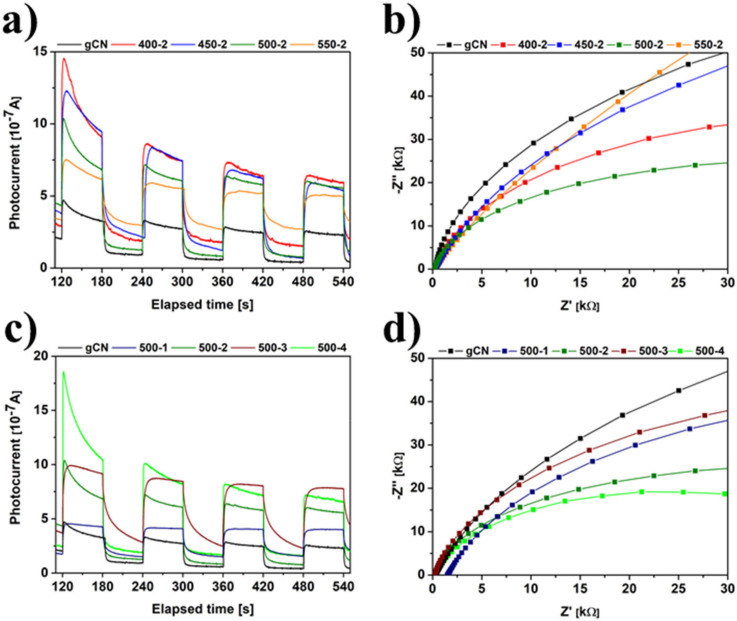
Chronoamperometry (CA) spectra of gCN prepared under different temperatures (**a**), Electrochemical Impedance Spectroscopy (EIS) spectra of gCN prepared at different temperatures (**b**), Chronoamperometry (CA) spectra of gCN prepared at different times (**c**), Electrochemical Impedance Spectroscopy (EIS) spectra of gCN prepared at different times (**d**).

**Figure 9 ijms-22-13096-f009:**
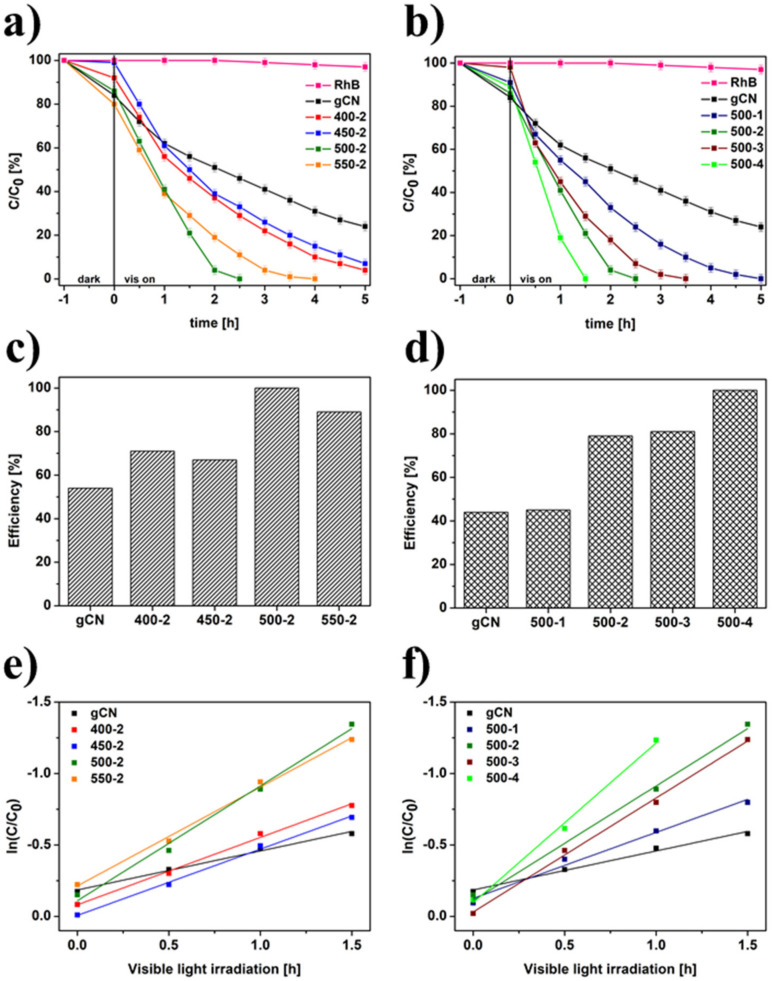
Photocatalytic degradation of RhB dye under visible light irradiation catalyzed by gCN prepared at different temperatures (**a**) and times (**b**), the efficiency of degradation of RhB dye under visible light irradiation catalyzed by gCN prepared at different temperatures (**c**) and times (**d**), the relationship between the RhB degradation efficiency and the light irradiation time for samples prepared at different temperatures (**e**) and times (**f**).

**Figure 10 ijms-22-13096-f010:**
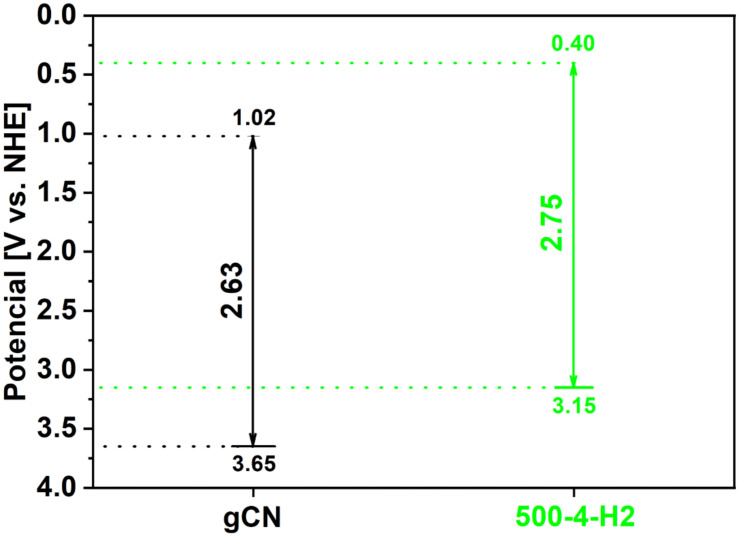
Schematic diagram of band structures (valance and conduction bands) of gCN and 500-4.

**Table 1 ijms-22-13096-t001:** BET data and energy bandgap values of gCN were prepared under different conditions.

Sample	Specific Surface Area [m^2^ g^−1^]	Total Pore Volume [cm^3^ g^−1^]	Average Pore Diameter [nm]	Energy Band Gap [eV]
gCN	8.89	0.003	0.88	2.63
400-2	30.33	0.014	1.05	2.65
450-2	16.81	0.008	1.35	2.68
500-1	14.76	0.007	1.00	2.69
500-2	28.69	0.013	1.03	2.70
500-3	21.45	0.010	1.29	2.71
500-4	24.11	0.012	1.09	2.75
550-2	25.72	0.012	1.25	2.81

**Table 2 ijms-22-13096-t002:** Preparation conditions of T-t samples.

Sample	Temperature [℃]	Duration [h]
400-2	400	2
450-2	450	2
500-1	500	1
500-2	500	2
500-3	500	3
500-4	500	4
550-2	550	2

## Data Availability

The data presented in this study are available online.

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
