# Peer review of "Influence of Hydrogenation on Morphology, Chemical Structure and Photocatalytic Efficiency of Graphitic Carbon Nitride"

_ijms, 2021, doi:10.3390/ijms222313096_

Round 1

Reviewer 1 Report

The manuscript by Baranowska et al. is investigating in detail how the annealing time and temperature influences the photoresponse and photocatalysis of g-CN. The manuscript has good structure, however some corrections are essential to be considered for publication:

  1. The authors claim that "annealing in H2 atmosphere causes the 122 hydrogenation of tri-s-triazine units forming dangling N – H moieties, breaking the weak 123 van der Waals forces between layers". Is there any explanation why in average the flakes in 500-4 are thicker than 500-2 rather than vice versa?
  2. In Figures 5-a,b the C-C/C=C have distinctive binding energies and should be separated
  3. "The highest photocurrent response is 245 achieved for 450-2". However in Figure 8-a the highest photoresponse is for the red colour (400-2) which also has a lower bandgap (2.63 over 2.65). More clarification should be provided here
  4. Is there any explanation why with higher temperature (550 oC) the efficiency for RhB degradation is reduced? (Figure 9c)
  5. Conclusion is very short. It should be expanded highlighting the most important results of this investigation. There is no mention of the 400-2 (or 450-2) photoresponse for example

Author Response

Point-by-point response to the reviewers’ comments on Ms. Ref. No.: IJMS-1477104

Response to Reviewer 1

The manuscript by Baranowska et al. is investigating in detail how the annealing time and temperature influences the photoresponse and photocatalysis of g-CN. The manuscript has good structure, however some corrections are essential to be considered for publication:

Response: We sincerely appreciate the constructive comments and suggestions, which are very helpful for us to improve the quality of the manuscript. The issues have been addressed point-by-point:

  1. The authors claim that "annealing in H2 atmosphere causes the hydrogenation of tri-s-triazine units forming dangling N – H moieties, breaking the weak van der Waals forces between layers". Is there any explanation why in average the flakes in 500-4 are thicker than 500-2 rather than vice versa?

Response: We would like to thank the Reviewer for paying attention to this issue. We calculated the mean thickness of 500-4 once again. Here, we calculated the mean value of thickness from a larger objects group of 500-4. The proper mean thickness of 500-4 is 2.28 nm. We corrected this in Table S1.

We would like to underline that, all 500-2, 500-3 and 500-4 are composed of 315 atomic layers. The obtained data confirm that longer time of H2 treatment (longer than 2h) did not lead to significant changes in the reduction of the thickness of the samples. It is due to that the exfoliation process is stabilized in 2h of H2 treatment at 500 °C, and the time extension at  500 °C did not significantly affect the parameters of the obtained samples.

  1. In Figures 5-a,b the C-C/C=C have distinctive binding energies and should be separated

Response: Thank you for this kind suggestion. Unfortunately, the data from XPS consist C-C/C=C are in one column, and we are not able to separate them correctly. Additionally, presented way of studies is commonly used by other researcher groups, eg. (1) Zhang, S., Li G., Duan L., Wang H., Zhao Y., Zhang Y. g-C3N4 synthesized from NH4SCN in a H2 atmosphere as a high performance photocatalyst for blue light-driven degradation of rhodamine B. RSC Adv. 2020 10, 19669-19685. https://doi.org/10.1039/D0RA02454F; (2) J. Mater. Chem., 2008, 18, 4893-4908 https://doi.org/10.1039/B800274F; (3) Qiao, Fengmin & Wang, Jiaomei & Ai, Shiyun & Li, Lifang. (2015). As a new peroxidase mimetics: The synthesis of selenium doped graphitic carbon nitride nanosheets and applications on colorimetric detection of H2O2 and xanthine. Sensors and Actuators B: Chemical. 216. http://dx.doi.org/10.1016/j.snb.2015.04.074.

  1. "The highest photocurrent response is achieved for 450-2". However in Figure 8-a the highest photoresponse is for the red colour (400-2) which also has a lower bandgap (2.63 over 2.65). More clarification should be provided here

Response: We would like to thank the Reviewer for this comment. The photocatalytic activity of 400-2 and 450-2 are similar as well as the photoresponse. To better understand the photoresponse of all studied samples we have corrected the units in Figure 8c from [10-8 A] to [10-7 A]. Thanks to this simple correction, which will be beneficial for the readers, it is much easier to choose a sample with the highest photoresponse.  

Moreover, to determine the most active sample, we consider also the photoresponse during light off (flat line). Considering different beginning levels, 450-2 has the highest photocurrent response (Figure 8a; to be more specific, the difference between 400-2 samples is 1 %). In Figure presented below, we present the procedure of measurement of photoresponse to make a suitable comparison.

  1. Is there any explanation why with higher temperature (550 oC) the efficiency for RhB degradation is reduced? (Figure 9c)

Response: Thank you for this suggestion. A short explanation of this phenomenon has been added in the main manuscript: “Increase in the temperature of H2 treatment up to 550 ℃ decreased removal efficiency of RhB. This is due to much higher value of band gap energy (2.81 eV), which influanced decrease in optical and electrochemical properties of 550-2. The data collected from photoluminescence, chronoamperometry and electrochemical impedance spectroscopy proves that the transfer and separation of photogenerated carriers are hampered in comparison to sample with the highest photoactivity (500-4) in the RhB degradation process”.

  1. Conclusion is very short. It should be expanded highlighting the most important results of this investigation. There is no mention of the 400-2 (or 450-2) photoresponse for example.

Response: Thank you for this suggestion. The Conclusion has been expanded including highlights the most important results of this investigation: “The improvement of  500-4 photoefficiency can be understood as the common impact of following parameters: reduction of thickness (from 6.46 to 2.28), increasing of specific surface area (from 8.89 to 24.11), reduction of PL intensity and enhancement of photoresponse”. We have mainly focused on 500-4 in this section because our main goal of this contribution was to optimize both parameters (temperature and time of annealing in the H2 atmosphere) in the photocatalytic RhB degradation process to achieve a highly effective photocatalyst.

Reviewer 2 Report

The authors presented a comprehensive work on the influence of hydrogenation temperature and time on the properties of graphitic carbon nitride including the morphology, chemical structure and photocatalytic. They found that annealing temperature at 500 degree for 4h in H2 atmosphere is critical to the photocatalytic activity, which is about 1.76 times higher than those without annealing. After detailled charcterization the underlying mechanism is due to creation of nitrogen vacancies on gCN surfaces which act as active sites and will improve the mobility of photogenerated charge carriers. This manuscript is well written and is very interesting to readers for photocatalytical research. It can be published in International Journal of Molecular Sciences after minor revision.

  1. It is difficult to find a clear trend for treatments at different temperatures. For example, the specific surface area and total pore volume for 400-2 is larger even compared to those treated in higher temperatures in Table 1. The PL intensity for 450-2 is very close to gCN, however 400-2 has distinct to gCN in Figure 7. If annealing temperature is thee cause of property change, why higher temperature leads to minor changes? What are the possible reasons for these results?
  2. The typos and formats in the whole manuscript need to be revised before publish, especially the units. For instance, Line 62, 68, 78 on page two.

Author Response

Point-by-point response to the reviewers’ comments on Ms. Ref. No.: IJMS-1477104

Response to Reviewer 2

The authors presented a comprehensive work on the influence of hydrogenation temperature and time on the properties of graphitic carbon nitride including the morphology, chemical structure and photocatalytic. They found that annealing temperature at 500 degree for 4h in H2 atmosphere is critical to the photocatalytic activity, which is about 1.76 times higher than those without annealing. After detailled charcterization the underlying mechanism is due to creation of nitrogen vacancies on gCN surfaces which act as active sites and will improve the mobility of photogenerated charge carriers. This manuscript is well written and is very interesting to readers for photocatalytical research. It can be published in International Journal of Molecular Sciences after minor revision.

Response: We appreciated the reviewer’s valuable comments and constructive suggestions, which are much beneficial to improving the scientific quality of this manuscript. We accepted all the suggestions and comments. According to the suggestions, a thorough revision has been performed by providing point-by-point response. We hope the revision could be satisfactory.

  1. It is difficult to find a clear trend for treatments at different temperatures. For example, the specific surface area and total pore volume for 400-2 is larger even compared to those treated in higher temperatures in Table 1. The PL intensity for 450-2 is very close to gCN, however 400-2 has distinct to gCN in Figure 7. If annealing temperature is thee cause of property change, why higher temperature leads to minor changes? What are the possible reasons for these results?

Response: We would like to thank the Reviewer for this comment. We agree that the clear trend for treatment at different temperatures is difficult to find. What is important is that the annealing of g-C3N4 in H2 atmosphere always improves both physicochemical and electrochemical properties of the studied samples, resulting in boosted photocatalytic efficiency. The common impact of parameters such as morphology (size, thickness), energy band gap and position of energy levels, specific surface area, life time of photogenerated carriers has a key role in photocatalysis. The answer to the question: “Why  higher temperature leads to minor changes?” is more complex than SSA and PL intensity only. Moreover, higher temperature leads to an increase in the value of Eg and a decrease size of gCN flakes. We believe that this is the explanation of the minor changes in the photocatalytic degradation process. The obtained results showed that the time of annealing in H2 has a crucial impact on photocatalytic efficiency, not the temperature of annealing. The main goal of this investigation was to optimize both parameters (temperature and time of annealing in H2 atmosphere) in the photocatalytic RhB degradation process to achieve a highly effective photocatalyst.

  1. The typos and formats in the whole manuscript need to be revised before publish, especially the units. For instance, Line 62, 68, 78 on page two.

Response: We would like to thank the Reviewer for pointing this issue. Appropriate changes have been made in the whole manuscript as suggested.
